# Harmful R-loops are prevented via different cell cycle-specific mechanisms

Marta San Martin-Alonso[1], María E. Soler-Oliva [1], María García-Rubio[1], Tatiana García-Muse [1✉] &
Andrés Aguilera [1✉]

Identifying how R-loops are generated is crucial to know how transcription compromises genome integrity. We show by genome-wide analysis of conditional yeast mutants that the THO transcription complex, prevents R-loop formation in G1 and S-phase, whereas the Sen1 DNA-RNA helicase prevents them only in S-phase. Interestingly, damage accumulates asymmetrically downstream of the replication fork in *sen1* cells but symmetrically in the *hpr1* THO mutant. Our results indicate that: R-loops form co-transcriptionally independently of DNA replication; that THO is a general and cell-cycle independent safeguard against R-loops, and that Sen1, in contrast to previously believed, is an S-phase-specific R-loop resolvase. These conclusions have important implications for the mechanism of R-loop formation and the role of other factors reported to affect on R-loop homeostasis.

[1] Centro Andaluz de Biología Molecular y Medicina Regenerativa CABIMER, Universidad de Sevilla-CSIC-UPO, Seville, Spain. ✉email: tatiana.muse@cabimer.es; aguilo@us.es

R-loops, formed by a DNA−RNA hybrid and the displaced single-stranded DNA (ssDNA), are a natural source of genome instability[1–3]. Numerous factors have emerged that protect cells from harmful R-loops. They can regulate R-loop levels by preventing their formation, a paradigmatic example being the THO complex, or by resolving them via RNA cleavage or DNA−RNA unwinding, as is the case of RNH1 and Sen1/SETX, respectively[2,3]. THO is critical for the formation of optimal mRNPs during transcription elongation and functions at the RNAPII-transcribed regions[4,5]. In yeast, this protein complex is formed by five interacting subunits: Tho2, Hpr1, Mft1, Thp2, and Tex1[6–8]. Their mutations cause transcription elongation and RNA export defects, and increase recombination and DNA damage in association with high levels of R-loops in yeast[9,10] and human cells[11]. Although evidence suggests that the main mechanism by which THO protects cells from R-loops relies on promoting an optimal assembly of an export-competent protein-coated mRNA, we now know that THO physically interacts with the Sin3A histone deacetylase and the UAP56/DDX39B DNA−RNA helicase, both of which also protect cells from R-loop accumulation[11,12]. Thus, THO may coordinate a complex co-transcriptional mechanism of DNA protection that relies on RNA packaging, localized chromatin compaction, and DNA-RNA unwinding. On the other hand, yeast Sen1 is an essential DNA−RNA helicase that participates in transcription termination of non-polyadenylated RNAs, including snoRNAs, snRNAs, and cryptic unstable transcripts[13]. sen1 mutants display high recombination rates associated with R-loops[14]. Interestingly, the human ortholog of Sen1, senataxin (SETX), a protein linked to DNA repair and neurodegeneration[15], functions in transcription termination and R-loop control in association with BRCA1[16,17]. Its DNA-RNA unwinding ability is required to remove hybrids formed at double-strand breaks (DSBs) to facilitate homologous-dependent repair (HDR)[18].

Despite the different factors protecting cells from harmful R-loops, the mechanism by which these are formed and cause genome instability is still unclear. Evidence indicates that nucleases such as XPG and XPF can cleave the ssDNA of an R-loop[19], but the main mechanism by which this leads to DNA damage is by blocking replication fork (RF) progression[20]. This has been shown in artificial systems[10,21], and in yeast and human cells depleted of R-loop suppressor factors such as THO, Sin3A, Fanconi Anemia, Sen1/SETX, the UAP56/DDX39B helicase, or the bromodomain chromatin factor BRD4[5,11,12,22–24]. Altogether, the data supports that R-loops associated with transcription-replication (T-R) conflicts are a source of DNA breaks, complying with the fact that transcription interferes with RF progression[25] and promotes genome instability[20,26,27]. We showed in the past that T-R collisions cause genome instability in head-on (HO) but not in co-directional (CD) orientation in yeast cells[28]. A study in human cells[29], extended the conclusions to an accumulation of R-loops seen at HO but not CD conflicts, leading to a model by which R-loops are formed at HO T-R conflicts. However, R-loops are detected in both HO and CD collisions when they are stabilized by an RNA binding factor, as it is the case of yeast Yra1[30]. Moreover, a recent genome-wide analysis shows no difference in R-loop signals at HO versus CD conflicts[31].

Identifying whether R-loops are caused by T-R conflicts or vice versa is crucial not only to decipher the mechanisms of R-loop formation, but also to know how R-loops compromise genome integrity. This acquires additional relevance after noticing that DNA breaks facilitate DNA−RNA hybrid accumulation[32], opening the possibility that breaks occurring at T-R conflicts could contribute to spontaneous R-loops. Here we demonstrate that R-loop formation is a co-transcriptional event that can occur independently of replication and that THO is a general safeguard against R-loop accumulation along the cell cycle. In contrast to previously believed, Sen1 is not a global DNA−RNA hybrid resolvase, but functions to remove DNA−RNA hybrids formed at T-R collision sites during S-phase. These conclusions have important implications for the mechanisms by which R-loop are originated and cause genome instability and invite to revisit our view on the role of factors reported to impact on R-loop homeostasis.

## Results

**Rapid harmful R-loop accumulation after Hpr1 and Sen1 depletion.** Aiming at deciphering the mechanisms by which cells minimize R-loop and/or DNA−RNA hybrid accumulation, we generated *hpr1* and *sen1* conditional mutants with the auxin-inducible degron (AID) system, *hpr1-aid,* and *sen1-aid*. Western-blot analysis of cell extracts at different time points after auxin addition, showed elimination of Hpr1 (after 60 min) and Sen1 (after 30 min) proteins (Fig. 1a). Serial dilution assays revealed that, in contrast to wild-type (WT), *hpr1Δ*, and *sen1-1* cells, both *hpr1-aid* and *sen1-aid* strains displayed a growth defect upon prompt Hpr1 or Sen1 depletion in the presence of 1 mM of auxin, a defect that was exacerbated in 2.5 mM (Supplementary Fig. 1a).

To know whether the two conditional mutants exhibited the genetic instability described for *hpr1Δ* and *sen1-1*[4,10,14], we measured Rad52-YFP foci as a readout of global DNA breaks[33]. Whereas auxin addition did not affect the percentage of WT cells showing Rad52 foci, both Hpr1 and Sen1 depletion significantly increased such a percentage (Fig. 1b and Supplementary Fig. 1b). Importantly, overexpression of RNase H1 fully suppressed this increase in Rad52 foci-containing cells after Hpr1 and Sen1 removal (Fig. 1b). Next, we assayed the effect on recombination using the *L-LacZ* system, consisting of direct repeats of two *leu2* alleles flanking the bacterial *lacZ* ORF that lead to *Leu+* recombinants by single-strand annealing (SSA). *Leu+* recombination frequencies increased in *hpr1-aid* and *sen1-aid* strains 20- and 10-fold after 1 mM auxin addition and 185- and 61-fold after 2.5 mM addition compared to the WT, which showed no significant changes with and without auxin (Fig. 1c). Overexpression of RNase H1 resulted in a significant reduction of this phenotype in *hpr1-aid* and *sen1-aid* strains (Fig. 1d), confirming that hyper-recombination relied on R-loops[9,14].

Next, we assessed whether this increase in DNA damage was linked to an increase in R-loops. This was assayed using the S9.6 antibody that recognizes DNA−RNA hybrids. Even though S9.6 detects DNA−RNA hybrids, regardless of whether or not part of an R-loop, studies with bisulfite mutagenesis suggest that in most cases hybrids correlate with R-loops[34–37]. Thus, here we will refer indistinctly to R-loops, even though in some cases the structure could be just a DNA−RNA hybrid. We performed immuno-fluorescence (IF) of chromosome spreads, to gain a broader view of DNA−RNA hybrid accumulation, and DNA−RNA hybrid immunoprecipitation (DRIP) to study the R-loop accumulation at specific genomic regions. First, chromosome spreads in asynchronous cultures after auxin-induced depletion revealed that cells with hybrids were 5% in the wild-type, but increased to 16% in *hpr1-aid* and to 19% in *sen1-aid* (Fig. 1e and Supplementary Fig. 1c). Both signal values were significantly reduced by in vivo RNase H1 overexpression (Fig. 1e and Supplementary Fig. 1c). Consistently, DRIP analysis after Hpr1 or Sen1 depletion in asynchronous cultures showed that both conditional mutants accumulated hybrids compared to WT at the *GCN4* and *PDR5* regions analyzed. The DRIP values were fully suppressed by in vitro RNase H1 treatment, confirming the specific detection of DNA−RNA hybrids (Fig. 1f and Supplementary Fig. 1d).

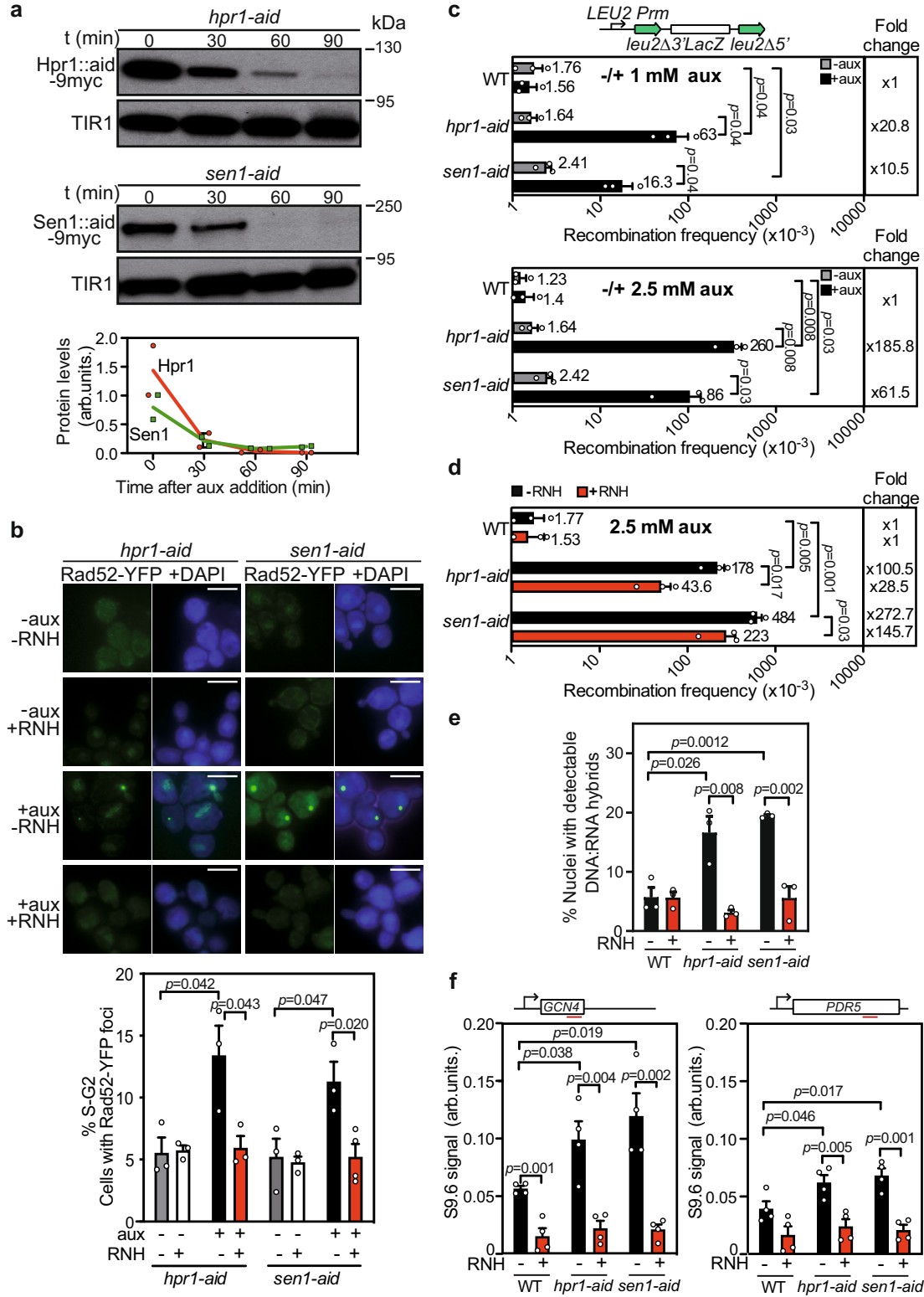

Altogether, these results prove that both proteins exert a direct role in preventing R-loop accumulation and R-loop-mediated recombinogenic DNA damage.

**Hpr1 and Sen1 depletion impairs replication and S-phase progression.** We next analyzed whether the growth defect caused by *sen1* and *hpr1* depletion was linked to a possible defect in cell cycle progression. Thus, we monitored cell cycle progression in WT, *hpr1-aid,* and *sen1-aid* cells in the presence or absence of auxin by fluorescence-activated cell sorting analyses (FACS) (Fig. 2a). Cells were grown up to an exponential phase, treated or not with auxin (to deplete Hpr1 or Sen1) and α-factor (to synchronize the cells in G1-phase) for 2 h, and then

**Fig. 1 Rapid Hpr1 or Sen1 depletion leads to R-loop-dependent genomic instability. a** Immunoblotting with αmyc after 0, 30, 60, or 90 min of 1 mM auxin treatment of *hpr1-aid* and *sen1-aid* cells (with aid-degron and myc translational fusions to the *HPR1* and *SEN1* genes, respectively). AtTIR1-9Myc was used as loading control. Quantification (mean) of protein levels normalized to TIR1 ($n = 2$ biologically independent samples). **b** Representative images and value of Rad52-YFP foci in *hpr1-aid* and *sen1-aid* strains with or without RNH1 and with or without auxin. Graph shows mean and SEM of independent experiments ($n = 3$). A minimum of 100 cells were counted per experiment. **c** Scheme of the *L-lacZ* direct-repeat recombination system. Recombination analysis in WT, *hpr1-aid,* and *sen1-aid* with and without the indicated auxin concentrations ($n = 3$ biologically independent experiments) **d** Recombination analysis in strains as in (**c**) with or without RNH1 in the presence of auxin ($n = 3$ biologically independent experiments). In (**c**, **d**), average and SEM are shown. **e** Percentage of positive DNA−RNA hybrids nuclei in chromosome spreads stained with the S9.6 antibody in WT, *hpr1-aid,* and *sen1-aid* asynchronous cells with or without RNH1. Data are presented as mean values $+/-$ SEM ($n = 100$ cells examined over three independent experiments). **f** DRIP analysis at *GCN4* and *PDR5* genes in cells as in (**e**), treated ($+$) or not ($-$) with RNH1 in vitro. Data are presented as mean values $+/-$ SEM ($n = 4$ biologically independent experiments). The *P* values were calculated by the two-tailed unpaired Student *t*-test. Scale bar 5 µm. Data underlying this figure are provided as Source data file.

washed and released into fresh medium with auxin and without α-factor. Wild-type and conditional mutants progressed through the cell cycle similarly under control conditions in the absence of auxin. On the contrary, upon Hpr1 depletion by auxin, *hpr1-aid* cells showed a clear delay in reaching the S/G2 phase. A similar delay in reaching the S/G2 phase was observed for the *sen1-aid* cells after Sen1-depletion compared to WT (Fig. 2a), consistent with previous data[22].

These delays in S-phase progression could be explained by a defect in replication. To test this possibility, we performed ChIP analyses of cells treated with BrdU, a thymidine analog that can be incorporated into replicating DNA and whose incorporation can be monitored by ChIP in properly modified yeast strains (Fig. 2b). We analyzed four regions where replication collides with transcription in a HO orientation, as validated in previous analysis[38], so that we could assay a proximal and a distal sequence with respect to each replication origin (ARS). A delay in BrdU incorporation was observed at the four regions tested in Hpr1-depleted cells (+aux) compared to the WT condition (−aux), demonstrating that replication is hampered in the absence of Hpr1 (Fig. 2b). In the case of *sen1-aid*, BrdU was incorporated in the two ARS proximal regions (1 and 3) with similar efficiency at early time points. However, in contrast to the WT condition (−aux) in which BrdU signals decrease, BrdU incorporation increases until it reaches a sustained plateau 10 min later (Fig. 2b). Instead, BrdU incorporation at the two ARS distal regions (2 and 4) was clearly delayed and less efficient (Fig. 2b). Altogether, these results demonstrate that depletion of either Hpr1 or Sen1 leads to replication impairment, but it seems to be of different nature in each case.

**Differential R-loop accumulation through the cell cycle**. Since a rapid depletion of Hpr1 or Sen1 leads to R-loop-dependent genetic instability and replication impairment of different nature we asked whether R-loops are accumulated differently during the cell cycle, which we monitored by FACS analyses (Supplementary Fig. 2a). First, we performed chromosome spreads and DRIP experiments in G1-synchronized cultures. Notably, whereas chromosome spreads of *hpr1-aid* cells showed a clear and statistically significant increase in the number of cells with S9.6 signal (approx. 4-fold), *sen1-aid* cells showed no significant differences with respect to WT (Fig. 3a and Supplementary Fig. 2b). DRIP analyses confirmed this result. Whereas Hpr1 depletion in G1-arrested cells increased R-loops with respect to the WT, Sen1 depletion did not (Fig. 3b and Supplementary Fig. 2c). Second, we determined R-loop accumulation in S-phase. For this, *hpr1-aid* and *sen1-aid* G1-arrested cells were released in fresh media to allow proliferation, and samples were collected in S-phase. As observed, the change of cell-cycle stage did not affect the basal S9.6 signal in WT cells. As in G1, depletion of Hpr1 exhibited a significant increase of R-loops compared to WT, as

detected by both IF of chromosome spreads and DRIP analysis (Fig. 3c, d and Supplementary Fig. 2b). Importantly, Sen1 depletion increased the percentage of cells with R-loop signal (approx. 7-fold), as determined by chromosome spreads (Fig. 3c and Supplementary Fig. 2b). This result was corroborated by DRIP-qPCR analysis, which showed that Sen1 depletion led to a significant DNA−RNA hybrid accumulation at the analyzed regions *GCN4* and *PDR5* (Fig. 3d).

These data unambiguously indicate that DNA−RNA hybrids increase during the S-phase upon Sen1 depletion but not during G1, whereas they increase during both G1 and S-phases upon Hpr1 depletion. This correlates with the observation that mRNA and protein levels of Hpr1 are detected throughout the cell cycle (Supplementary Fig. 2d, e). We asked next whether DNA−RNA hybrids formed in S-phase, as is the case of *sen1-aid* cells, could pass unresolved to the following G1-phase. To test this, we performed a new experiment in which G1-synchronized cells were allowed to complete one cell cycle, and then we measured R-loops in the following G1-phase (hereafter called G1$_2$). As observed in Fig. 3e, R-loop levels detected by DRIP in *hpr1-aid* and *sen1-aid* cells in the two regions studied, were higher than in the WT in G1$_2$. Thus, R-loops formed at one stage of the cell cycle could pass unresolved to another stage and to the next cell cycle.

**R-loop-accumulating mutants show increased DNA damage through the cell cycle**. Although removal of Hpr1 or Sen1 leads to R-loop accumulation in a cell cycle phase-specific manner, both conditional mutants accumulate Rad52 foci. Therefore, we examined how R-loops associate with DNA damage in relation to cell cycle phases using phosphorylation of histone H2A at S129 (H2AP mark) as a readout of DNA damage[39]. We first assayed H2AP levels by western blot, using protein extracts obtained from cultures in G1, S, and G1$_2$-phases. In G1, only Hpr1 depletion exhibited elevated H2AP levels, whereas in S-phase depletion of either Hpr1 or Sen1 increased H2AP levels above WT (Fig. 4a, b). Again, in the G1$_2$ phase, only Hpr1 and not Sen1 depletion showed an increase of H2AP with respect to WT (Fig. 4a, b).

To assay whether the high DNA damage detected was linked to R-loops, we assessed the presence of H2AP by ChIP in S-phase cultures overexpressing or not RNase H1. Rapid depletion of both Hpr1 and Sen1 significantly increased the H2AP signal at the *GCN4* and *PDC1* genes. Importantly, the signal was reduced, even though slightly, upon RNase H1 overexpression (Fig. 4c). Therefore, since R-loops formed in G1 or S lead to DNA damage, the ability of a harmful R-loop to result in DNA damage is independent on the cell cycle stage at which it is formed.

**Loss of Hpr1 and Sen1 lead to differential genome-wide DNA−RNA hybrid accumulation in S-phase**. Next, we wondered how R-loops and DNA damage association occurs genome-wide,

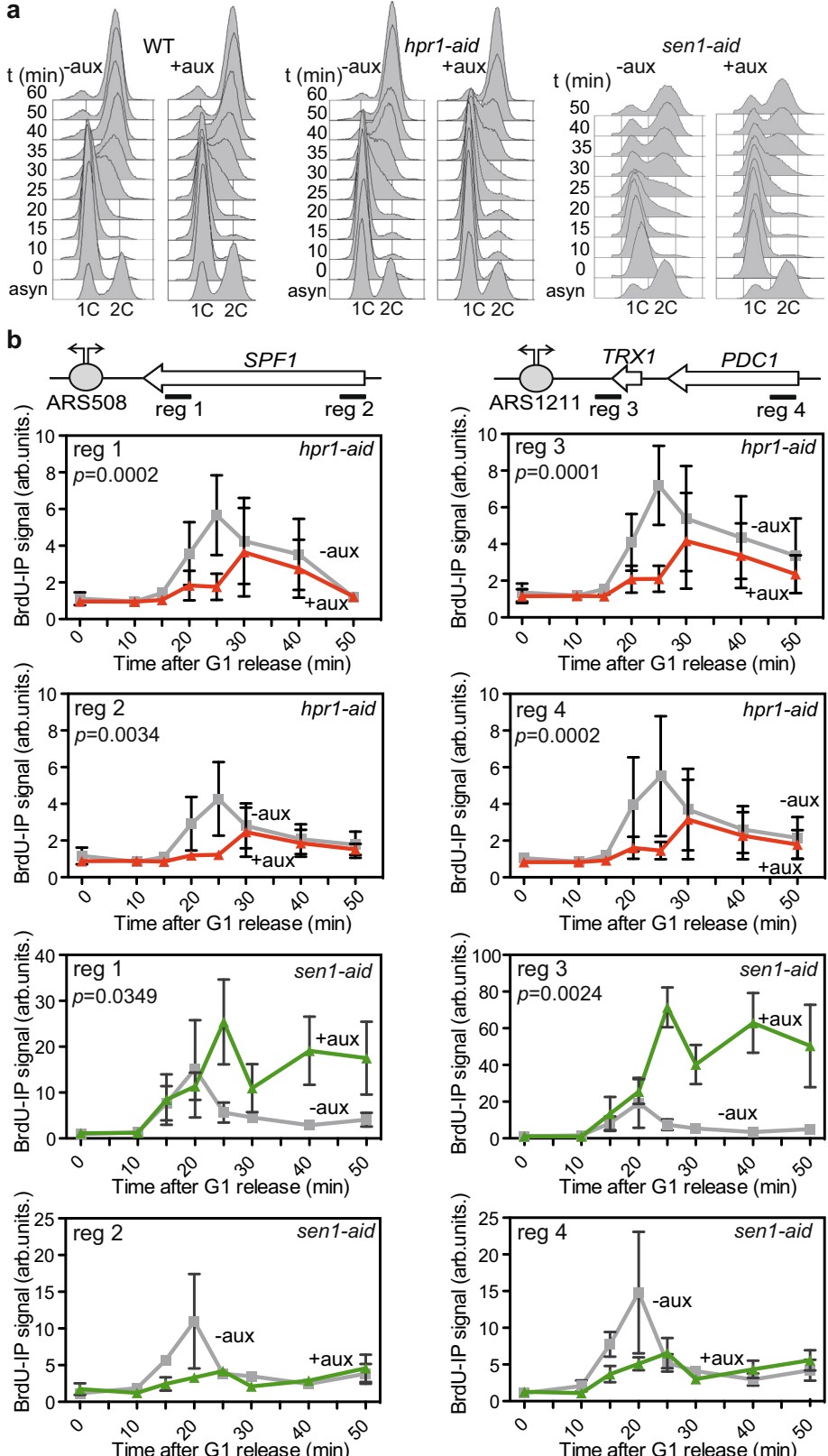

**Fig. 2 Quick depletion of Hpr1 or Sen1 impairs replication. a** Analysis of cell cycle progression after alpha factor release of WT, *hpr1-aid,* and *sen1-aid* strains with and without auxin by FACS. **b** ChIP analysis of BrdU incorporation into the DNA at the genomic regions and amplicons depicted in the schemes (top) in the *hpr1-aid* and *sen1-aid* degron strains with and without auxin. Mean $+/-$ SEM is shown ($n = 3$ biologically independent experiments). The $P$ values calculated by the two-tailed Wilcoxon signed-rank test are shown for each condition.

for which we focused on S-phase at which DNA−RNA hybrids can be detected in both mutants. To obtain the genome-wide R-loop distribution after depletion of Hpr1 or Sen1 in S-phase synchronized cells, we conducted DRIPc-seq[40]. We implemented this method by adding an RNase III treatment prior to S9.6 immunoprecipitation that permits the elimination of dsRNA, which is also recognized by S9.6[41,42]. Genomic DNA−RNA hybrid profiles of both *hpr1-aid* and *sen1-aid* cells in S-phase show regions with DNA−RNA hybrids, and the overexpression of RNase H leads to undetectable levels of DNA−RNA hybrids establishing that the immunoprecipitated material corresponded to DNA−RNA hybrids (Supplementary Fig. 3a). Two replicates of DRIPc-seq were performed and were highly correlated with each other (Supplementary Fig. 3b).

The genome-wide coverage of R-loop peaks was obtained using chromstaR[43]. Two representative genomic regions are shown in Fig. 5a. DNA−RNA hybrids accumulate along the whole genome in the WT and two conditional mutants, but both *hpr1-aid* and *sen1-aid* cells exhibit a peak enrichment (referred to peaks with higher levels than WT, whether or not pre-existing) of 5.7- and 4.3-times over the WT, respectively (Supplementary Fig. 3c). The data indicate that Hpr1 and Sen1 depletion causes a general R-loop-enrichment genome-wide. Accordingly, metaplot analyses of all enriched peaks show a higher DRIPc-seq signal for *hpr1-aid* and *sen1-aid* strains with respect to WT (Fig. 5b). Analysis of DRIPc-seq signal locations revealed that 85% of *hpr1-aid* and 78% of *sen1-aid* DRIPc-seq peaks coincide with protein-coding genes (Supplementary Fig. 4a), in agreement with a role of both proteins in RNAPII-driven transcription[4,13]. Comparison of R-loops formed by sense and anti-sense transcripts reveals that DNA−RNA hybrid accumulates in both strands at gene bodies, but with a progressive increase towards the 3′ region with a peak at the transcription termination site (TTS) in the sense strand that is not observed in the anti-sense orientation apart form a sharp peak coincident at the TTS (Fig. 5c). It is worth noting that genes showing increased DNA−RNA hybrids in the mutants are longer and expressed at higher levels than the genome average (Supplementary Fig. 3d). In addition, metaplot analysis reveals a specific increase over the WT at telomeres in *hrp1-aid* mutant not observed in *sen1-aid* (Fig. 5d), whereas at tRNAs and snoRNAs the increase is observed in *sen1-aid* and not in *hpr1-aid* (Fig. 5e and Supplementary Fig. 4a), consistent with their previously reported functions[44,45]. In addition, the volcano plot of *hpr1-aid* and *sen1-aid* peaks reveals a 17.3 and 23% of specific peaks, respectively (Supplementary Fig. 3c). Accordingly, the metaplot analysis of the peaks specific to *hpr1* and *sen1* shows higher DRIPc-seq signals (Supplementary Fig. 4b) and, in both mutants, shows a clear enrichment at gene bodies of both sense and anti-sense signals (Fig. 5f). For common peaks, the increase is similar for both mutants and also observed preferentially at gene bodies (Fig. 5f and Supplementary Fig. 4b). Altogether, these genomic data indicate that both proteins have a global but distinct role in R-loop control.

Depletion of Hpr1 in G1-arrested cells increased R-loops with respect to the WT; therefore, we assessed if the S-phase DNA−RNA hybrids observed in *hpr1-aid* cells were originated in G1 or de novo. DRIPc-seq in WT and Hpr1-depleted G1-synchronized cultures revealed a genome-wide coverage of DNA−RNA hybrid peaks in both strains along the whole genome. Representative regions showing DRIPc-seq peaks of both DNA strands in G1 in *hpr1-aid* and WT (Supplementary Fig. 5a) or the joined data of both strands compared between G1 and S-phase are shown (Supplementary Fig. 5b). Consistent with the DRIP-qPCR results, R-loops were differentially enriched in the *PDC1*, *PDR5*, and *GCN4* regions in G1 and S phase (Supplementary Fig. 5c). There is a 2.7-fold enrichment of G1 peaks in *hpr1-aid* cells over the WT (Supplementary Fig. 5d), pointing out to a widespread R-loop-enrichment also in G1 phase. Interestingly,

after Hpr1 depletion, 75% of total peaks (1053 out of 1462) detected in G1-phase are observed in S-phase (Supplementary Fig. 5e, left), but 90% (347 out of 387) of the G1 *hpr1-aid* R-loop-gain peaks (those significantly enriched in G1 over the WT), are not enriched in S-phase (Supplementary Fig. 5e, right). Instead, 2051 new R-loop-gain peaks are detected in S-phase. Notably, metaplot analysis of all these peaks shows that *hpr1-aid* R-loop-gain peaks detected in G1 (clearly enriched in G1 over the WT in S-phase) either disappear (G1-specific, Supplementary Fig. 5f) or, if still enriched, the levels are much lower (G1-S common) (Supplementary Fig. 5g). Importantly, the level of *hpr1-aid* R-loop-gain peaks detected in S-phase (Supplementary Fig. 5h) is much lower than that of G1-enriched peaks in G1-phase (compare Supplementary Fig. 5h, bottom with Supplementary Fig. 5f, g, top). Altogether, the data indicate that Hpr1 protects from R-loops at both G1 and S phase, but its effect is much stronger during G1 phase.

**S-phase R-loops correlate with specific sites of DNA damage.** Provided the differential impact on R-loop accumulation at G1 and S-phase and at distinct genomic regions, we wondered how this different R-loop control affects DNA damage. We extended the S-phase genome-wide analysis with a ChIP-seq of H2AP. We found a background level of H2AP along the genome in WT cells consistent with S-phase replication being the most vulnerable DNA process and therefore prone to DNA breaks (Supplementary Fig. 6a). Such enrichment of H2AP was also observed in both *hpr1-aid* and *sen1-aid* strains, with signal intensities at specific sites clearly above WT levels, *sen1-aid* exhibiting the highest signals. When subtracting the WT signals from those from each conditional mutant to focus on the signals specifically enriched after Hpr1 or Sen1 depletion, a differential group of regions sensitive to H2AP accumulation in both mutants is observed (Supplementary Fig. 6b).

Next, we did a bioinformatic comparison between the S-phase DNA−RNA hybrid (DRIPc-seq) and H2AP (ChIP-seq) genome-wide profiles for each strain considering the H2AP signal average of the 5 kb region around each R-loop peak. While the R-loops of WT cells show low H2AP signal in those 5 kb regions, both mutants show a significant H2AP accumulation spreading from the DNA−RNA hybrid sites (Fig. 6a, b). When we restricted the analysis of the H2AP profile to regions surrounding enriched DRIPc-seq peaks for each mutant, we found that in both *hpr1-aid* and *sen1-aid* mutants H2AP occupancy was higher than in the WT, as expected (Fig. 6c and Supplementary Fig. 6c). Remarkably, whereas this occupancy was symmetric toward each side of the R-loop peak in *hpr1-aid*, it was asymmetric in *sen1-aid* (Fig. 6c), a feature that can be seen even better around the *sen1-aid* specific R-loop sites (Supplementary Fig. 6c).

Since it has been previously observed in triple mutants *rnh1 rnh2 sen1* that persistent DNA−RNA hybrids show asymmetric recruitment of the DSB repair factor Rad52, we extended our comparative analysis to the available data from Rad52 ChIP-seq[46]. Metaplot analysis of Rad52 recruitment to DRIPc-seq peaks in WT cells did not show the pattern of H2AP, suggesting that only a fraction of sites detected by H2AP in WT cells may correspond to breaks that are repaired via Rad52 (Supplementary Fig. 6d). However, *sen1-aid* cells show a similar asymmetric pattern of Rad52 signal distribution around the *sen1-aid* specific R-loop peaks as the H2AP signal (Supplementary Fig. 6d, e), which confirms that the increase in H2AP reflects mainly DNA breaks that can be repaired via Rad52. Therefore, the difference in the pattern of H2AP symmetry between *hpr1-aid* and *sen1-aid* mutants must reflect a profound difference in the way R-loops generated in G1 (*hpr1-aid*) and S-phase (*sen1-aid*) impact on DNA break formation and repair.

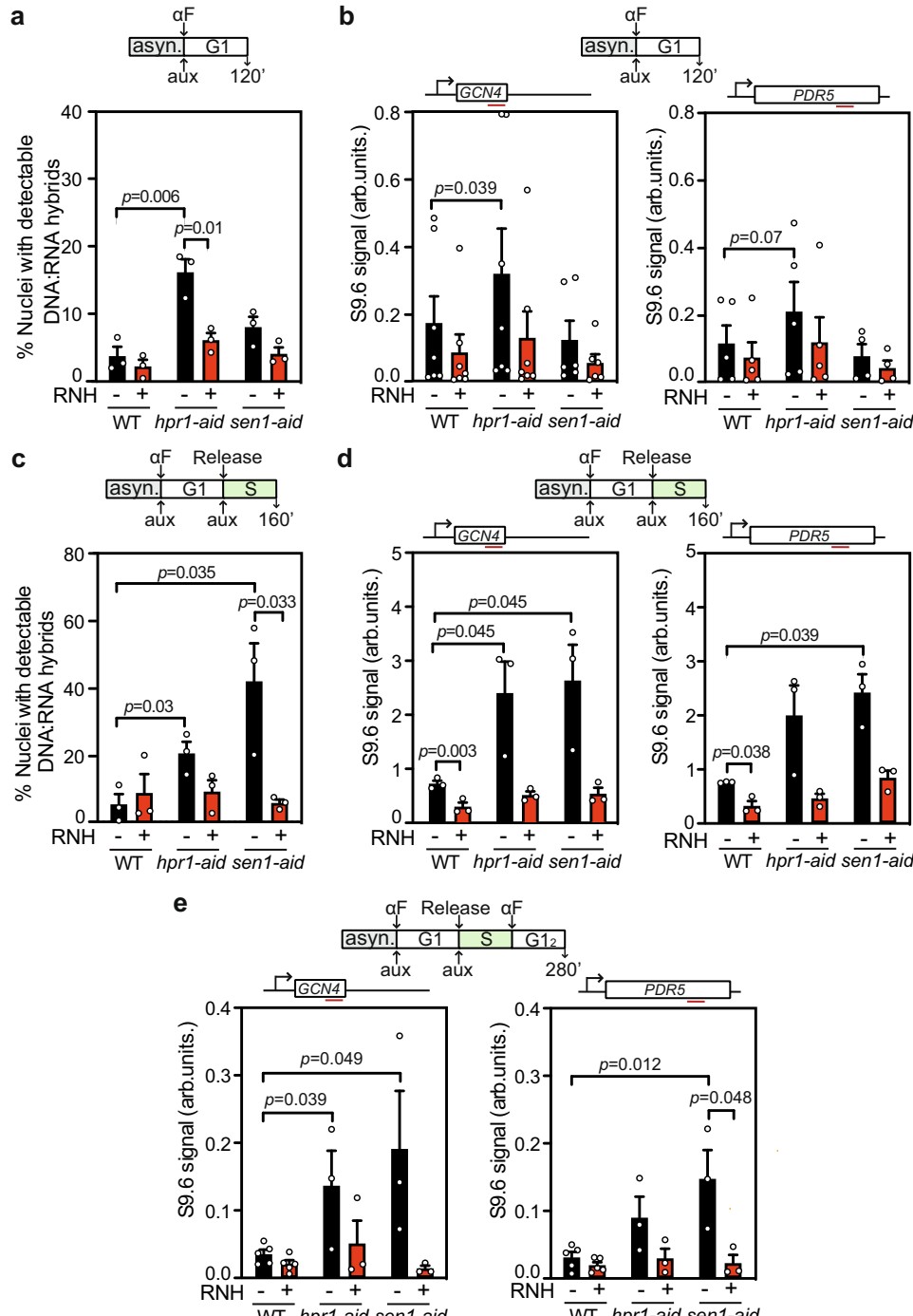

**Fig. 3 Hpr1 or Sen1 depletion causes nuclear DNA−RNA hybrid accumulation differently through the cell cycle. a** Percentage of positive nuclei for DNA −RNA hybrids in chromosome spreads stained with the S9.6 antibody in G1-phase WT, *hpr1-aid,* and *sen1-aid* cells with or without RNH1. Data are presented as mean values +/− SEM ($n=100$ cells examined over three independent experiments). **b** DRIP with S9.6 antibody at *GCN4* and *PDR5* genes in G1-phase WT, *hpr1-aid,* and *sen1-aid* cells treated (+) or not (−) with RNH1 in vitro. Data are presented as mean values +/− SEM ($n=6$ and $n=4$ biologically independent experiments for *GCN4* and *PDR5*, respectively). **c** Percentage of positive nuclei for DNA−RNA hybrids in chromosome spreads stained with the S9.6 antibody in S-phase cells as in (**a**). Data are presented as mean values +/− SEM ($n=100$ cells examined over three independent experiments). **d** DRIP with S9.6 antibody at *GCN4* and *PDR5* genes in S-phase cells as in (**b**), treated (+) or not (−) with RNH1 in vitro. Data are presented as mean values +/− SEM ($n=3$ biologically independent experiments). **e** DRIP with S9.6 antibody at *GCN4* and *PDR5* genes in G1-phase after one complete cell cycle (G1$_2$) cells as in (**b**), treated (+) or not (−) with RNH1 in vitro. Data are presented as mean values +/− SEM ($n=3$ biologically independent experiments). The *P* values were calculated by the two-tailed unpaired Student *t*-test. A diagram of the experiment is represented above each plot.

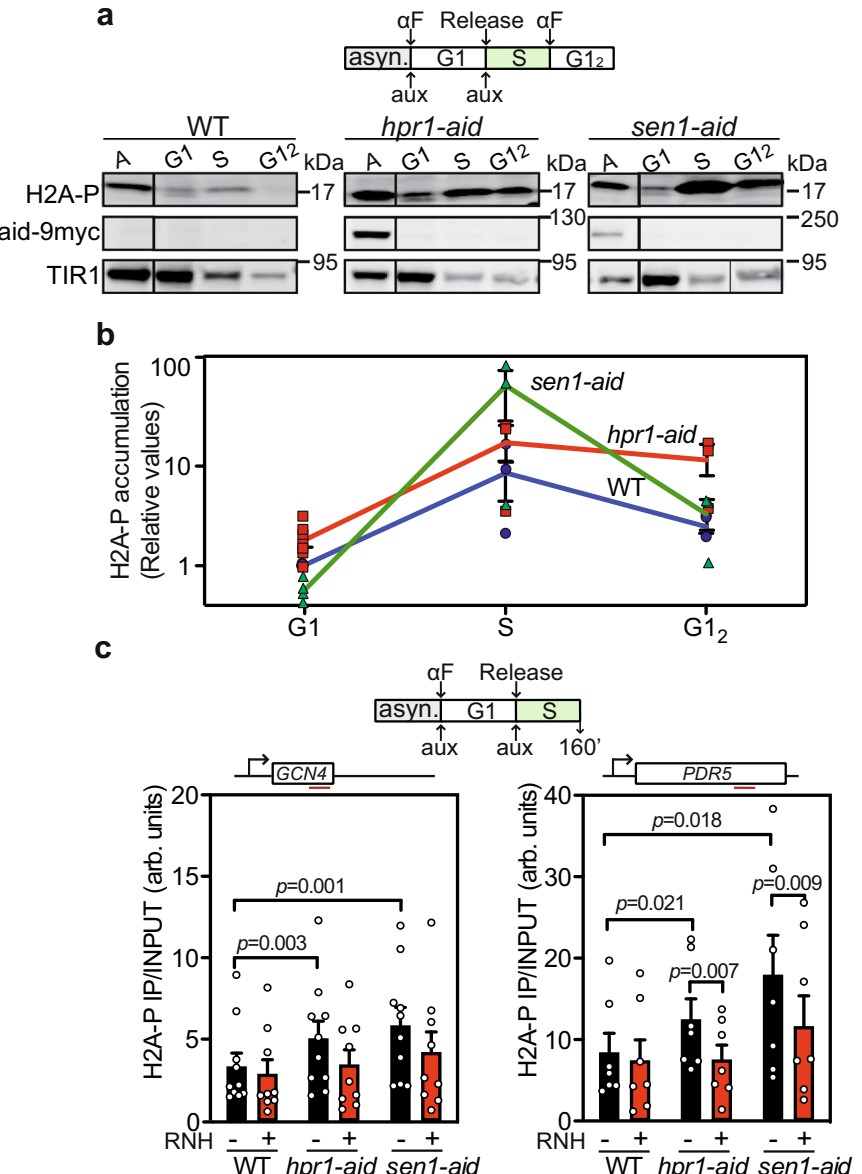

**Fig. 4 Hpr1 and Sen1 degron strains accumulate H2AP in S-phase. a** Immunoblotting with αH2AP of WT, *hpr1-aid,* and *sen1-aid* cells in asynchronous (A), G1, S-phase, and G1-phase after one complete cell cycle (G1₂) cultures. AtTIR1-9Myc was used as loading control. **b** Immunoblotting quantification (means +/− SEM) of protein levels normalized to the WT ($n = 3$ biologically independent samples). **c** H2AP ChIP analysis at the *GCN4* and *PDC1* genes in WT, *hpr1-aid,* and *sen1-aid* strains with or without RNH1. Data are presented as mean values +/− SEM ($n = 7$ biologically independent experiments). The *P* values are calculated by the two-tailed unpaired Student *t*-test. A diagram of the experiment is represented above each experiment.

Finally, we wonder whether this difference in replication defects in both mutants could be extended to regions containing replication origins (ARSs) (Supplementary Fig. 7a). Metaplots of H2AP signal distribution at early ARSs show a remarkable increased signal in the *sen1-aid* strain (Fig. 6d). The genes near early ARSs showed an increased H2AP signal in the *sen1-aid* mutant regardless of whether they were oriented HO or CD versus the RF while the *hpr1-aid* mutant did not show such an increase with respect to the WT (Supplementary Fig. 7b). This result suggests that transcription in the absence of Sen1 promotes T-R conflicts more prone to cause DNA breaks than in the absence of Hpr1. To address if the orientation in T-R conflicts is relevant in our conditional mutants, we grouped the ARS-proximal genes depending on their CD or HO orientation and calculated the percentage of those genes with common and S-phase *hpr1-* or *sen1-* specific R-loops. While the proportion of

ARS-proximal genes that accumulate R-loops in *hpr1* cells (either specific or common with *sen1*) is similar between HO and CD orientation, the genes that accumulate *sen1*-specific R-loops are mainly in the HO orientation (Fig. 6e). The results confirm the different nature and consequences of the R-loops prevented by each factor.

## Discussion

THO is the paradigm of conserved factors that prevent R-loop formation[9]. If inactive, a suboptimal protein-coated nascent mRNA is generated prone to hybridize back with the DNA template. However, human THO works together the mSin3A deacetylase complex and the UAP56 DNA−RNA unwinding activity to prevent R-loop accumulation behind an ongoing RNAPII[11,12], and inactivation of THO, mSin3A, and UAP56/

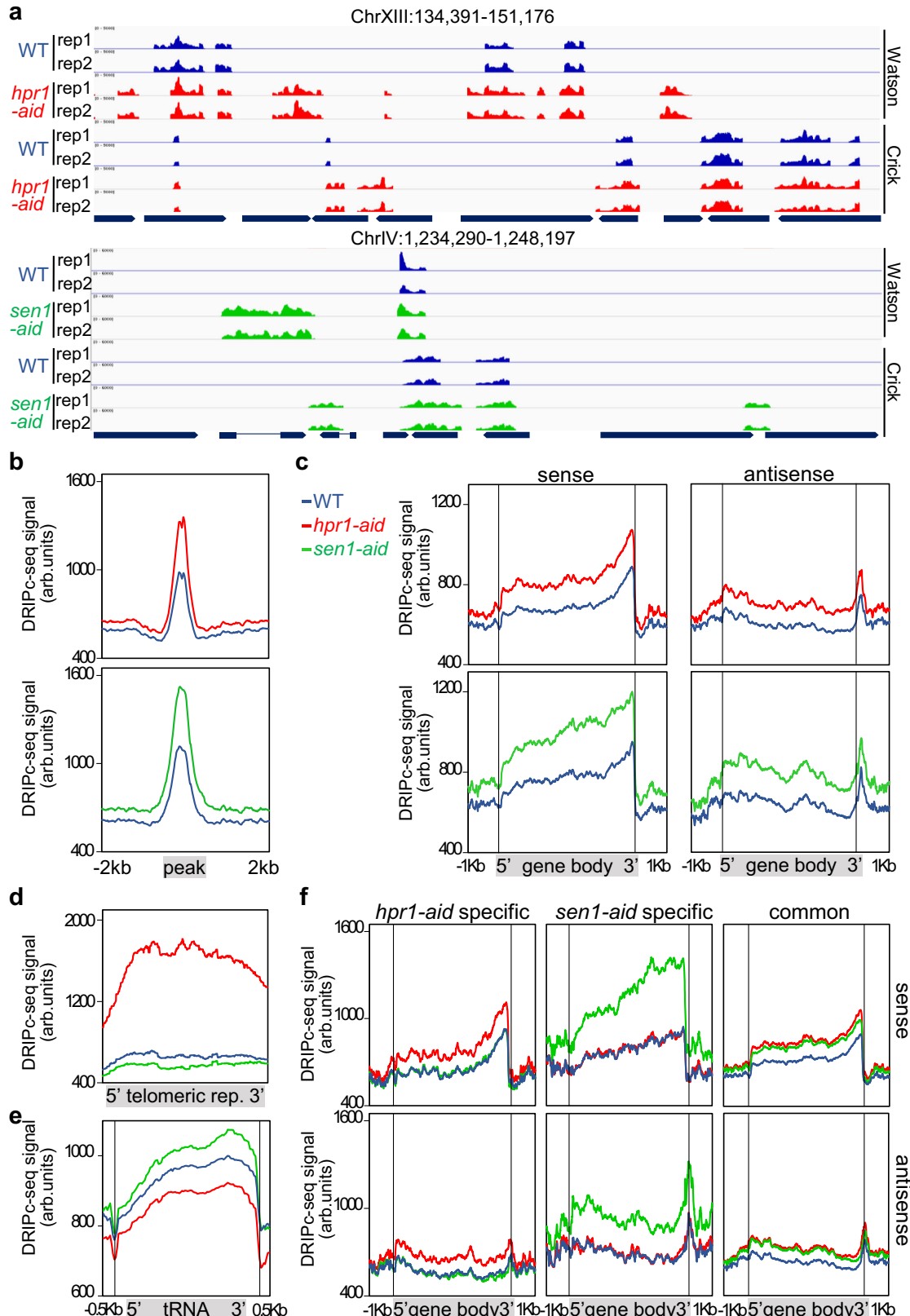

**Fig. 5 DNA−RNA hybrids genome-wide accumulation after Hpr1 and Sen1 controlled depletion. a** Representative screenshots of different genomic regions showing the DRIPc-seq signal of detected peaks profiles for WT (blue), *hpr1-aid* (red), and *sen1-aid* (green) mapped at Watson and Crick strand (*n* = 2). **b** DRIPc-seq signal (average coverage) metaplot analysis of peaks ±2 Kb enriched in samples as in (**a**). **c** Distribution of antisense and sense DRIPc-seq signal (average coverage) along a gene metaplot for cells as in (**a**). **d** Distribution of DRIPc-seq signal (average coverage) at telomeric repeats metaplot in samples as in (**a**). **e** Distribution of DRIPc-seq signal (average coverage) at tRNAs metaplot in samples as in (**a**). **f** Distribution of antisense and sense DRIPc-seq signal (average coverage) along specific enriched genes metaplot in samples as in (**a**).

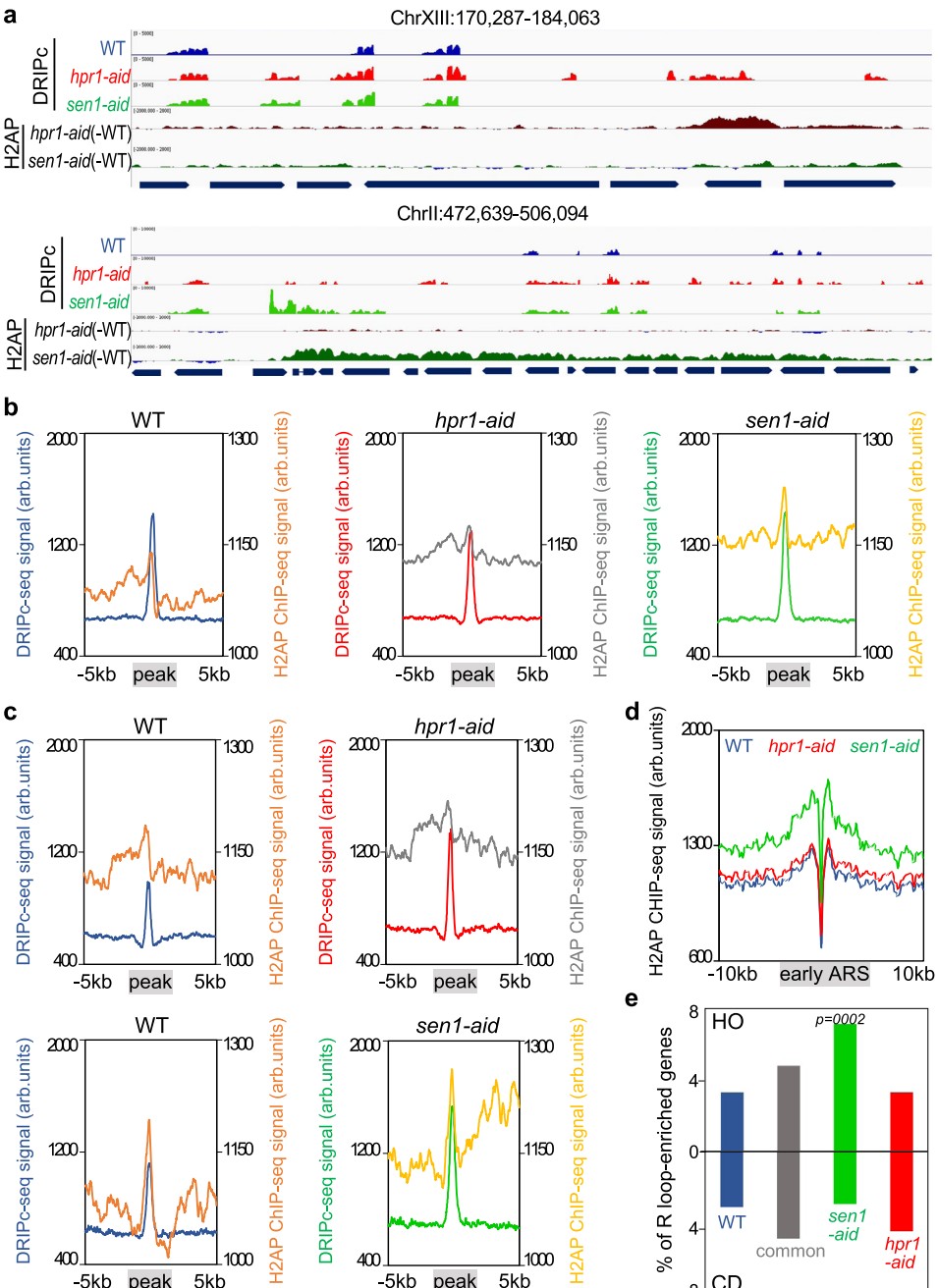

**Fig. 6 H2AP genome-wide accumulation after Hpr1 and Sen1 controlled depletion correlate with DNA−RNA hybrids. a** Representative screenshots of different genomic regions in which DRIPc-seq signal of detected peaks (average coverage *n* = 2) from WT (blue), *hpr1-aid* (red), and *sen1-aid* (green) cells. The H2AP ChIP-seq signal from *hpr1-aid* (dark red) and *sen1-aid* (dark green) subtracting the WT signal is also shown. **b** Metaplot analysis of DRIPc-seq peaks ±5 Kb and H2AP ChIP-seq signal detected in WT (orange), *hpr1-aid* (gray), and *sen1-aid* (yellow) cells. Other details as in (**a**). **c** Metaplot analysis of DRIPc-seq peaks ±5 Kb and H2AP ChIP-seq signal of specific peaks in *hpr1-aid* cells and *sen1-aid* cells with respect to the WT. Other details as in (**b**). **d** Distribution of H2AP ChIP-seq signal (average coverage) along early ARS ± 10 Kb from WT (blue), *hpr1-aid* (red), and *sen1-aid* (green) cells. **e** Percentage of genes R-loop-enriched in WT (blue), *hpr1-aid* specific (red), *sen1-aid* specific (green), and common enriched (gray) that would collide with the replication forks coming from early ARS in head-on (top) or codirectional (bottom) orientation. The *P* values were calculated by the chi-square test one-sided, one degree of freedom.

DSDX39B in yeast and/or human cells results in R-loops that block RF progression leading to DNA breaks, as has been shown also for other chromatin and RNA processing factors[10,21,24,38,46–49]. Importantly, here we report that THO inactivation in synchronized cells clearly shows that R-loops accumulated at high levels in G1-arrested cells and after entering S-phase (Fig. 3). Even though the genome-wide analysis reveals that additional R-loops are enriched in S-phase in *hpr1-aid* cells,

the enhancement is clearly below that observed in G1 (Supplementary Fig. 5f, h). This result suggests that no S-phase replication-dependent mechanism contributes to R-loop accumulation in THO mutants in G1. Therefore, transcription and RNA processing constitute a relevant source of harmful R-loops regardless of T-R conflicts. On the other hand, Sen1 is considered a paradigm of DNA−RNA hybrid unwinding resolvase[50,51]. The involvement of Sen1/SETX in R-loop control has been shown in

yeast and human cells for both the spontaneous origin of DNA damage and the repair of DSBs[14,16–18,22,46,52–54], leading to the view that Sen1/SETX it is a master key in R-loop homeostasis. However, Sen1/SETX is involved in different processes from transcription termination[16,17,55,56] to DNA repair related to neurodegenerative ataxia[57–61]. Importantly, in contrast to *hpr1-aid*, cells depleted of Sen1 only accumulate R-loops to a detectable level once they entered S-phase (Fig. 3). The results indicate that Sen1 does not have a major role in protecting from harmful R-loops in G1, consistently with the recent observation that Sen1 only acts in S-phase[62]. Therefore, Sen1, in contrast to previously believed, may not be a general and major DNA−RNA helicase able to act on all types of hybrids.

The comparative analysis of the genome-wide impact of depleting Hpr1 and Sen1 during S-phase reveals clear differences in the origin of R-loops and their link to genome instability (Fig. 5). As expected, there is a high correlation between the regions accumulating R-loops above WT levels and those accumulating DNA breaks, as determined by H2AP, in both *hpr1-aid* and *sen1-aid* mutants (Fig. 6). In both cases, R-loops compromise genome integrity during S-phase. The results are consistent with previous data showing that R-loops or DNA−RNA hybrids, no matter their origin, are associated with RF blockage putatively responsible for DNA breakage and genome instability[20]. They agree also with the fact that SETX colocalizes with 53BP1 and other DNA damage response (DDR) proteins in response to replication arrest[61] and can function at DSBs formed at transcriptionally active loci[18]. In this sense, it is worth noticing that it is likely that DNA−RNA hybrids controlled by the THO complex, whether in G1 or S phase, correspond to R loops, consistent with previous bisulfite mutagenesis analyses[36]. However, a remaining question is whether in S-phase, and in particular in *sen1-aid* cells, DNA-RNA hybrids may also form during replication with a non-replicated lagging strand without generating an R-loop[63,64].

R-loops accumulate at HO and CD T-R conflict sites with similar ratio in *hpr1* cells, and DNA breaks, as inferred from H2AP signals, accumulate symmetrically around the R-loop-enriched regions with equal distribution downstream and upstream of the RF (Fig. 6). This together with the high accumulation of R-loops in G1 and S cells supports that in THO mutants a major proportion of highly increased co-transcriptional R-loops are the cause of RF blockage and not the result of T-R conflicts. Thus, we conclude that R-loops may form by co-transcriptional failures independently of replication, and for which eukaryotic cells have developed the transcription-associated machinery, a paradigm of which is the THO complex, to prevent them[9]. In this sense, the larger impact of HO T-R conflicts as a source of DNA breaks and replication impairment compared to CD conflicts was shown in yeast cells[28]. Interestingly, in human cells, only HO conflicts were associated with R-loops[29], but in yeast cells overexpression of the RNA binding protein Yra1, able to bind DNA−RNA hybrids, also increased R-loops at CD conflicts[30], consistent with the conclusion that R-loops in WT cells form regardless of the orientation of the T-R conflict, even though those enriched at CD collisions would be more efficiently removed.

Strikingly, depletion of Sen1 has a very different outcome. Not only R-loops are formed preferentially in S phase, but they associate at a higher ratio with HO than with CD T-R conflict sites (Fig. 6). In addition, there is a clear asymmetric distribution of H2AP damage around the R-loop peaks. This result supports the notion that R-loops can form at a higher ratio at HO T-R conflict sites, but a specific machinery that includes the Sen1 resolvase would prevent them, so that in WT cells with an active Sen1 such an asymmetry would not be observed. A large

difference between HO and CD conflicts is the supercoil accumulated between the two advancing machineries[31,47,65]. Interestingly, using asynchronous cultures of a conditional *sen1* mutant, Rad52 was observed to accumulate ahead of the RF in the HO orientation at specific sites[46]. Although this observation does not exclude the possibility that even with breaks occurring at similar ratio in HO and CD conflicts, only those ahead of the RF at HO T-R conflicts would be repaired by Rad52-mediated HDR, it reinforces the conclusion that R-loops accumulate preferentially at sites of HO T-R conflicts when Sen1 is depleted. It would be interesting to explore in the future whether this HO preference is related to their specific topological constraints, provided that in CD conflicts the negative supercoil upstream of the RNAP would be counteracted with the positive supercoil generated ahead of the approaching RF, whereas HO conflicts will result in accumulation of positive supercoil between the RF and the transcription machinery[20].

It is possible that the positive supercoiled DNA, accumulated ahead of the RNAPII, promotes its rotation facilitating the dsDNA unwinding behind the RNAPII and thus DNA−RNA hybrids formation (Fig. 7). Since Sen1 is a helicase involved in transcription termination, which has been shown to play a key role at RF encounters with TTSs[22,66], and function in S-phase[62], it may be crucial to remove such R-loops and to prevent major consequences of the RF blockage. Consistently, pausing at the TTS of highly expressed human genes containing R-loops prevents HO T-R conflicts[67]. Alternatively, Sen1, and by extension SETX, might have a specific role in the HO T-R conflicts if these are more prone to generate DSBs, which could be catalyzed by Top2 given its preferential action of positive supercoiled DNA, but also via Slx4 as recently has been shown to participate in T-R conflicts during S-phase[68]. Formation of DSBs at HO T-R conflicts would facilitate hybridization of RNA with DNA, being a preferential target for Sen1 as has been shown for SETX at DSBs in transcribed regions[18]. The specific mechanisms by which Sen1 would be a major player in the resolution of R-loops at HO T-R conflicts would need to be investigated.

In summary, our study reveals that R-loops can be formed by different mechanisms at distinct stages of the cell cycle, consistent with the recent observation suggesting that they can be of diverse nature given their capacity to induce different DDRs in human

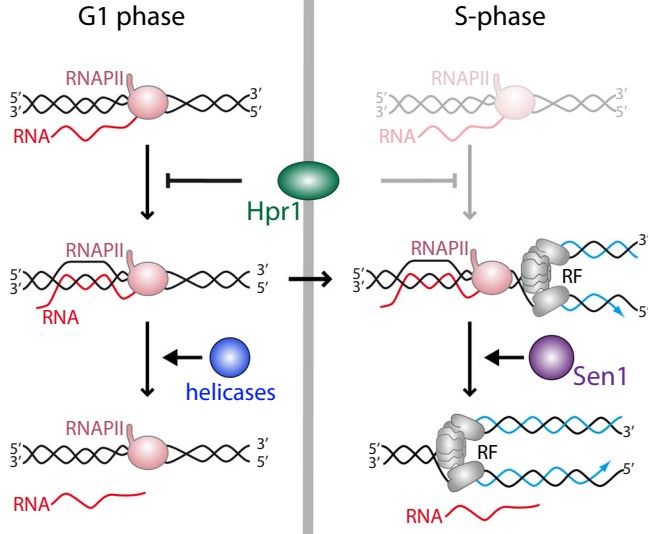

**Fig. 7 Model.** During G1 and S-phase co-transcriptional R-loops are prevented by the THO complex, whereas S-phase R-loops induced by T-R conflicts are resolved by the Sen1 DNA−RNA helicase.

cells[63]. Thus, specific factors are in charge of protecting cells from their accumulation at different cell cycle stages. As indicated in Fig. 7, R-loops may form in G1, regardless of replication, in which co-transcriptional factors like THO would play a key role in preventing them. Yet, if formed, DNA−RNA helicases, like the THO interactor UAP56/Sub2[8,11] or the yeast homologs of some of the DDX human family reported to unwind DNA−RNA hybrids[2], would be able to remove them; but if not, and R loops remain in S phase, they would block RF progression leading to replication stress, DNA damage and genome instability. Instead, in S phase, apart from co-transcriptional prevention mechanisms, HO T-R conflicts would preferentially promote accumulation of hybrids that are removed by the Sen1 helicase, which would specifically eliminate hybrids at T-R conflicts. Thus, a major mechanism of R-loop prevention would be independent of replication and would rely on the THO complex, whereas another one would be linked to T-R conflicts and would rely on the Sen1 DNA−RNA helicase.

## Methods

**Yeast strains and media**. Yeast strains used in this study are derivatives of W303 (*MATa his3-11,15 leu2-3,112 Δtrp1 ura3-1, ade2-1 can1-100*) and are listed in Supplementary Table 1.

Degron yeast strains used in this study were generated after integration of the construct pHyg−AID*−9myc purified from a parental plasmid pSM409 in the C-terminal of the proteins Hpr1 or Sen1 in cells in TIR1 expressing cells YMK612[69].

Media used in this study: YPAD (1% yeast extract, 2% bacto-peptone, 2% glucose, and 20 mg/ml adenine), Synthetic defined (SD) (0,17% yeast nitrogen base (YNB) without amino acids, 0,5% ammonium sulfate, supplemented with amino acids). The absence of amino acid/s is specified when required, synthetic complete (SC) (SD with 2% glucose), Sgal (SC with 2% galactose), Sraff (SC with 2% raffinose), SGL (SC with 3% glycerol and 2% sodium lactate). Sporulation medium (SPO) (1% potassium acetate, 0.1% yeast extract, 0.005% glucose). Solid media was prepared adding 2% agar before autoclaving.

For serial dilution growth assays, mid-log cultures were grown in YPAD medium. Ten-fold serial dilutions of the culture were prepared with sterile water and 3 μl of each dilution was spotted on plates. These were incubated for 2–3 days at 26 °C.

For cell cycle analysis mid-log cultures were synchronized at G1 with α-factor (biomedal). The arrest was confirmed by microscopic observation after 120 min. Release from arrest was achieved by two washes with fresh medium and addition of pronase (Sigma) to a final concentration of 50 μg/μl.

For protein depletion, auxin (3-indole acetic acid, IAA) (Sigma) was diluted to a final concentration of 1 mM. Unless indicated, all experiments were carried out 2 h after auxin addition.

Yeast strains were defrosted from glycerol stocks and grown at 30 °C except for *sen1-1* cells that were grown at 26 °C, using standard practices.

**Genetic analysis of recombination**. Recombination frequencies were calculated as the median value of six independent colonies. The average of three independent transformants was plotted. For the *LlacZ* system[4], yeasts were grown in SC-trp plates and Leu+ recombinants were selected in SC-leu-trp.

**DNA−RNA hybrid immunoprecipitation (DRIP)**. Asynchronous, G1-synchronized or S-phase mid-log cultures growing in YPAD or synthetic medium were collected and treated as in ref. [70] with variations. Briefly, after incubation with 1% of sodium azide (100 ml for DRIP qPCR and 400 ml for DRIPc-seq), each 50 ml of culture were washed twice with chilled water and the pellets were resuspended in 1.4 ml of spheroplasting buffer in 2 ml tubes (1 M sorbitol, 10 mM EDTA pH 8, 0,1% ß-mercaptoethanol, and 2 mg/ml Zymoliase 20T) and incubated during 30 min at 30 °C (50 rpm shaking). The pellets were rinsed with bidistilled water and resuspended with 1 ml G2 buffer (800 mM guanidine HCl, 30 mM Tris-HCl pH 8, 30 mM EDTA pH 8, 5% tween-20, and 0.5% Triton X-100). The samples were incubated at 37 °C after adding 40 μl of RNAse A (10 mg/ml) to remove ssRNA. Then, the samples were incubated with 75 μl proteinase K (20 mg/ml) 1 h at 50 °C. After centrifugation of the samples for 10 min at 7000 rpm, DNA was extracted with chloroform−isoamyl alcohol 24:1. To precipitate DNA, the samples were gently mixed with isopropanol and the DNA was spooled on a glass rod, washed with 70% ethanol, and incubated for 20 min at RT. DNA samples were resuspended in TE 1X and digested overnight with an enzyme cocktail containing EcoRI, XbaI, HindIII, BrsGI, and SspI (New England Biolabs) (with RNase III in order to remove dsRNA). For the negative control, half of the DNA was treated with RNase H (New England Biolabs) overnight at 37 °C.

For immunoprecipitation, 30 μl Dynabeads Protein A (Thermo Fisher) were incubated overnight with 3 μl of the S9.6 antibody per sample rotating at 4 °C in binding buffer (10 mM NaPO4 pH 7, 140 mM NaCl, and 0.05% Triton X-100) in TE. Each sample was incubated for 2 h at 4 °C rotating at low speed with 30 μl of the complex beads-S9.6 in a final volume of 500 μl binding buffer. Beads were washed three times with binding buffer and eluted in 120 μl elution buffer (50 mM Tris pH 8, 10 mM EDTA, and 0.5% SDS). Finally, samples were incubated for 45 min with 7 μl proteinase K at 55 °C and purified with the Nucleospin gel and PCR clean-up Macherey-Nagel purification kit. Real-time quantitative PCR was performed using iTaq universal SYBR green (Biorad) with a 7500 Real-Time PCR machine (Applied Biosystems).

**Genome-wide samples**. For DRIPc seq after the immunoprecipitation, samples were treated with DNAse I, and the RNA was purified with RNeasy mini kit (QIAGEN). The resulting RNA was subjected to library construction using the NEBNext ultra II Directional RNA Library prep kit for Illumina (NEB) from the fragmentation step. Then, samples were sequenced on the Illumina platform NextSeq500.

For H2A-P ChIP seq after chromatin immunoprecipitation and purification, DNA was subjected to library construction using ThruPLEX DNA-Seq 6S (12rxn) kit for Illumina. Then, samples were sequenced on the Illumina platform NextSeq500.

**DRIPc-seq, ChIP-seq, and RNA-seq read mapping, peak calling, annotation, comparison, and visualization**. Sequenced paired-ends reads were subjected to a quality control pipeline using the FASTQ Toolkit V.1.0.0 software (Illumina) and then mapped to the *Saccharomyces cerevisiae* reference genome sacCer3 using the Rsubread V2.0.1 software package with unique=TRUE parameter[71]. For DRIPc-seq, mapped reads were assigned to Watson or Crick strand using SAMtools V1.10[72]. Peak calling on DRIPc-seq data was performed with chromstaR V1.12.0 software package[43]. A multivariate analysis considering only peaks present in the two replicates (S-phase experiments) and with a value of maximum posterior in the peak cut-off of 0.99 999 was performed. Peaks smaller than 100 bp were discarded. For comparative analysis, regions covered by peaks in the two conditions that are being compared were merged and fused when closer than 200 bp distance using BEDtools V2.27.1[73]. The differential enrichment of these regions in each condition was performed using csaw V1.20.0 software package[74]. First we counted count reads in full genome using windowCounts() with bin = TRUE and width = 200 parameters. Then normalization factors were calculated using normOffsets(). After that, estimateDisp(), glmQLFit() and glmLRT() from edgeR package (v3.20.9), was used in order to calculate log2FC and *p*-value of the peaks. R-loop enriched regions were established selecting those peaks whose DRIPc signal fold change was higher than 1.2 X (2X for G1 data sets) and the—log 10 (*p*-value) was higher than 0.6 (1 for G1 data sets). After that, R-loop enriched regions in *hpr1-aid* and *sen1-aid* conditions were merged and fused again when closer than 200 bp distance using BEDtools[73]. The differential enrichment of these regions in each condition was performed the same way as before and R-loop enriched regions were divided in "*hpr1-aid* specific", "*sen1-aid* specific" and "common" according to the same criteria as before. In order to compare G1 and S phase conditions, since they differ in the number of replicas, we checked if R-loop-gain peaks detected in each condition overlap with one another, defining three categories: "G1 R-loop-gain peaks", "G1-S R-loop-gain peaks", and "S R-loop-gain peaks". In the S-phase analysis, enriched and specific peaks were annotated to the genomic features retrieved from the *Saccharomyces Genome Database*[75] where they overlap or were closer than 200 bp upstream using ChIPpeakAnno V3.28.1 software package[76], allowing each peak to be annotated to more than one genomic feature.

Comparison of WT expression levels and R-loop-enriched genes. RNA-seq data from ref. [77] was used. Samples were mapped to the *Saccharomyces cerevisiae* reference genome sacCer3 using the Rsubread V2.0.1 software package with unique=TRUE parameter[71]. RPKM normalization method was performed using windowCounts() tool from csaw software package[74].

Coverage profiling of ChIP-seq and DRIPc-seq were obtained using bamCoverage tool from deepTools V3.4.3[78]. Coverage plots were represented in the 5′ to 3′ direction. A bin size of 10 and normalization by RPKM were used. DRIPc plots show the average signal of the two replicates, if applicable. In order to compare DRIPc plots with Rad52 ChIP-seq signal, data from ref. [46] were used.

Genome example regions were plotted using IGV V2.8.2 software[79]. The background was removed from DRIPc tracks, so only regions considered as peaks were plotted. Also, WT H2AP ChIP-seq signal is subtracted from the mutants signal. In order to compare properly the samples treated or not with RNH1, we applied a scale factor to RNH-treated samples based on the ratio between the uniquely mapped reads and the total reads of each sample. Early ARS coordinates from ref. [80] were used. Only protein-coding genes located closer than 1 kb from the ARS midpoint were considered. These genes were split into codirectional or head-on genes according to their orientation with respect to the fork's orientation. Then, the percentage of these genes, which were R-loop enriched in each condition, was plotted.

**Miscellanea**. Analysis of yeast growth, Western, Rad52-YFP foci detection, fluorescence-activated cell sorting analyses (FACS) using a FACScalibur Becton Dickinson machine (settings as shown in Supplementary Fig. 8 using BD CellQuest

Pro), BrdU incorporation[38], Chromatin immunoprecipitation (ChIP)[38], Chromosome spreads immunofluorescence[81] and yeast cultures were performed using standard procedures. Western blots bands were quantified using ImageStudio software (LI-COR biosciences). For image acquisition, Leica fluorescence microscope DM6000B (AF6000) with a 100× objective and processed using the LAS AF software (Leica) and Adobe Photoshop. Primers used are listed in Supplementary Tables 2 and 3. Statistical analyses were performed using GraphPad Prism Software. Unless indicated, a two-tailed Student's test was performed in all the experiments. The number of experiments (n) and p values are indicated in the figure legends or in the figures. Plasmids used in this study are listed in Supplementary Table 4. The antibodies used are described in Supplementary Table 5.

**Reporting summary**. Further information on research design is available in the Nature Research Reporting Summary linked to this article.

## Data availability

The DRIPc-seq and H2AP ChIP-seq data generated are available at NCBI's Sequence Read Archive (SRA) under accession number GSE159870. Genomic features coordinates used are available at Saccharomyces Genome Database (SGD) [https://www.yeastgenome.org/]. RNA-seq data and Rad52 ChIP-seq data used are available at NCBI's Sequence Read Archive (SRA) under accession number SAMN11070697 and GSE110575, respectively. Early ARS coordinates are available at DOI: 10.1186/1471-2164-15-791. All data is available from the source data and/or the authors upon reasonable request. Source data are provided with this paper.

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

## Acknowledgements

This work was supported by grants from Spanish Ministry of Economy and Competitiveness (BFU2016-75058-P), European Research Council (ERC) Advanced Investigator Grant (ERC2014 AdG669898 TARLOOP), and the European Union (FEDER). M.S.M.-A. and M.E.S.-O. were holders of predoctoral training grants from the Spanish Ministry of Education, Culture and Sport and the Spanish Ministry of Economy and Competitiveness, respectively.

## Author contributions

T.G.-M. and A.A. designed the project and wrote the paper. M.S. M.-A. performed most of the experiments. M.G.-R. performed the DRIPc-seq. M.E.S.-O. performed the bioinformatics analysis.

## Competing interests

The authors declare no competing interests.
