## [Peer Review File · Nature Communications]

REVIEWER COMMENTS

Reviewer #1 (Remarks to the Author):

In this manuscript, Aguilera and colleagues nicely show that, in contrast to what is commonly hypothesized, the co-transcriptional R loop formation is independent of DNA replication. Using auxin-inducible degron system to control expression of the THO complex component Hpr1 and the Sen1 helicase, they provide convincing evidences for a role of the THO complex in preventing R-loop formation in both G1 and S phase whereas Sen1 resolvase function is restricted to S-phase. Genome-wide analysis clearly confirm the role of R-loops in induction of genome instability. The work and conclusions presented here are supported by clear data resulting from appropriate experimental approaches. Such a piece of work would undoubtedly be of interest for the field of gene expression and genome integrity and to my opinion, deserves publication after the following concerns being addressed.

Major comments

1. Genome-wide R-loop distribution in the *hpr1-aid* should be analyzed in both G1 and S phase in order to precise the role of Hpr1 on R loop control along the cell cycle and affirm that "no S-phase replication-dependent mechanism contributes to R loop accumulation in THO mutants apart from that of G1» as suggested in the discussion.
2. Sen1 expression is low in G1 and early S and strongly increases in late S and G2/M, as clearly shown by Mischo et al (ref.63). This explains why no R-loop accumulation is observed in G1 upon Sen1 depletion. This should be clear mentioned early in the text. On the same line, Hpr1 expression along the cell cycle should be analyzed.

Minor comments

1. Beside Figure 1a, what are the experimental conditions used for auxin treatment (concentration, time of treatment)?
2. In Figure 1b, images are not representative as Rad52 foci seem more frequent in *sen1-aid* cells in contrast to the quantification. What does mean n? Number of experiments (and then how many cells per experiment) or number of cells?
3. Figure 4 is far to be convincing. Adding more experiment would improve the statistical relevance of this experiment.
4. 25 out of 57 references (37%) correspond to self-citations. Although Aguilera's lab is undoubtedly a pioneer and leader in R-loop understanding, it is maybe exaggerated.
5. Methods section could be more detailed. For exemple, there is confusion between SC and SD, pronase treatment is not detailed...

Reviewer #2 (Remarks to the Author):

The Aguilera has led the field in defining the effects of R-loops on genome stability using yeast as a model system. They previously defined factors that prevent (THO complex) and remove (Sen1 RNA-DNA helicase) R loops and showed that head on (HO) conflicts between transcription and replication (T-R) conflicts are more detrimental than are co-directional (CD) conflicts. The current study used an auxin-inducible degron (AID) system to examine the cell cycle dependent effects of Hpr1 (THO component) and Sen1 loss, with a focus on establishing whether R loops cause or are the result of T-R conflicts. Loss of either protein elevated R-loop accumulation (chromatin spreads and DRIP at GCN4 and PDR5), strongly stimulated recombination and caused delays in S-phase progression. A key experiment was the use of synchronized cells to demonstrate that loss of Hpr1 was associated with R-loop accumulation throughout the cell cycle (R loop formation does not require T-R conflicts), while loss of Sen1 was associated with R-loop accumulation in S phase. Metaplot analyses of genomic data (R loops by DRIPc-seq; H2AP and Rad52 by CHIP-seq)

corroborated the distinct roles on Hpr1 and Sen1 on R-loop accumulation, with HO-CD difference in T-R conflicts being evident only in the *sen1-AID* strain. Altogether, the data convincingly demonstrate distinct and unexpected cell cycle-specific differences in R-loop accumulation and consequences upon loss of Hpr1 versus Sen1. The absence of THO results in co-transcriptional R-loops in G1 that cause subsequent replication fork blockage in S phase, while Sen1 loss only has effects in S phase and is associated with a HO bias for R loops as well as an asymmetry in damage. This is an important study that will fundamentally change how the field relates R loops to genetic instability.

Major comments

1. The model needs a bit more explanation. What are the non-Sen1 helicases that remove co-transcriptional R-loops? How does this model explain the observed HO bias in yeast? Doesn't the model predict that the HO bias should only be evident in the absence of Sen1? Or, is there something that preferentially deals with CD conflicts that occur outside of S phase? Lines 351-352 imply that something is "more efficiently" removing R loops at CD collisions. Although not required, directly examining how the HO bias is affected by Hpr1 versus Sen1 loss would be a welcome experiment.
2. The English needs editing throughout.

Minor comments

1. The authors need to define the THO complex components at its first mention.
2. In Fig 1b the *sen1-AID/+aux/-RNH* panels for Rad52-YFP and DAPI are identical.
3. Fig 1d – too many significant figures for -fold effects.
4. Fig S1c – why do the chromatin spreads (DAPI staining) look so different from WT?
5. Lines 152-156: there is no region 1 or 2 (should be 3' and 5'?) depicted in Fig 2b. Also, the WT BrdU data should be shown for comparison in Fig 2b, especially since auxin has subtle effects on the cell cycle in Fig 2a.
6. Any idea why the BrdU signal increases in the proximal regions (relative to WT) in the *sen1-aid* strain?
7. Line 165 – the ">16%" is misleading (implies only a 16% increase in the signal) and should be changed "from 4% to 16%" or "~ 4-fold." Same comment for line 174.
8. Fig 4a – define again what TIR1 is.
9. Line 197 – why "again" when *hpr1-aid* and *sen1-aid* were the same with respect to persistent Rad52 foci in G1?
10. Fig 5a – peak "enrichment" in text implies the amplification of pre-existing peaks. The data shown, however, do not convey an increase in peak signals but rather the appearance of new peaks.
11. Line 240 does not make sense.
12. Why is there an enrichment of anti-sense as well as sense signals in Fig 5c?

Reviewer #3 (Remarks to the Author):

The paper by Martin-Alonso et al, aims at elucidating the mechanism(s) that prevent R-loop formation/accumulation at different cell cycle stages. They used *hpr1* and *sen1* conditional mutants, that are known to accumulate R-loops, and analysed R-loop accumulation at different cell cycle stages. They found that while *hpr1* mutants accumulate R-loops in G1 and S phase, Sen1 prevents R-loop accumulation, specifically in S phase. The most important message here is that R-loops can form co-transcriptionally, and independently of DNA replication.

Specific Comments:

1. RnaseH overexpression in *hpr1* and *sen1* cells decreases recombination frequencies marginally but significantly (Fig1D). However, percentage of Nuclei with detectable hybrids decreases to wild type levels following RnaseH expression (Fig1E and also 1F). Question is, how much is recombination really depended on R-loops? Authors should discuss this point.

2. Authors selected 2 regions which are believed to have head-on collisions between forks and transcription and used BrdU-ChIP, to measure fork movement with respect to time. Due to differences in the replicates (3 qPCR triplicates), it is difficult to visualize the differences between +Aux and -Aux. Surely, when you take average you do see the differences, but is this significant?
3. (Brambati et al) showed that in *sen1* mutants besides, ARS1211, another dormant origin is activated (ARS1211.5). This is ignored in the paper and should be commented. It is formally possible that higher signals seen in *sen1* derive from the activation of ARS1211.5 (region 4).
4. The difference in H2A-P chip between -RNH and +RNH is not really significant at GCN4 but significant at PDC1. This needs a comment.

General comments:

The field of R-loop formation and accumulation is rather confusing. Particularly, it is generally believed that R-loops are always genotoxic and result from the uncoordinated clash between replication and transcription. This paper contributes to clarify an important point and convincingly demonstrate that R-loops can form independently of replication. Overall, the paper is solid and the results convincing. However, I strongly encourage the authors to clarify in the text that the R-loop concept is an interpretation as they monitor the presence of RNA-DNA hybrids, not R-loops. In fact, I would rather prefer that throughout the results section they refer to RNA-DNA hybrids and then, in the discussion, they clarify this issue. For instance, it is possible that in G1 the hybrids are in a R-loop conformation while in S phase they are not due to replication of the non transcribed strand.

REVIEWER COMMENTS

Reviewer #1 (Remarks to the Author):

In this manuscript, Aguilera and colleagues nicely show that, in contrast to what is commonly hypothesized, the co-transcriptional R loop formation is independent of DNA replication. Using auxin-inducible degron system to control expression of the THO complex component Hpr1 and the Sen1 helicase, they provide convincing evidences for a role of the THO complex in preventing R-loop formation in both G1 and S phase whereas Sen1 resolvase function is restricted to S-phase. Genome-wide analysis clearly confirm the role of R-loops in induction of genome instability. The work and conclusions presented here are supported by clear data resulting from appropriate experimental approaches. Such a piece of work would undoubtedly be of interest for the field of gene expression and genome integrity and to my opinion, deserves publication after the following concerns being addressed.

Thanks very much for the positive reception of the manuscript and the constructive comments and suggestions.

Major comments

1. **Genome-wide R-loop distribution in the *hpr1-aid* should be analyzed in both G1 and S phase** in order to precise the role of Hpr1 on R loop control along the cell cycle and affirm that “no S-phase replication-dependent mechanism contributes to R loop accumulation in THO mutants apart from that of G1» as suggested in the discussion.

We have performed a DRIPc-seq in G1 arrested WT and *hpr1-aid* cells as suggested, and the new data have been analyzed and compared to those of S-phase. R loops can be seen in both phases but the increase levels observed in G1 is much higher than in S phase. New hybrids are also observed in S phase, consistent with the global role of the THO complex in transcription regardless of cell cycle phase, and we have considered the possibility that in *hpr1-aid* cells R loops could also form at replication, even though at a lower level. We have included this analysis in the new Figure S5. The new results are explained in page 9 (lines 310-331), and have been discussed appropriately in pag. 12 (lines 424-428).

2. Sen1 expression is low in G1 and early S and strongly increases in late S and G2/M, as clearly shown by Mischo et al (ref.63). This explains why no R-loop accumulation is observed in G1 upon Sen1 depletion. This should be clear mentioned early in the text. On the same line, Hpr1 expression along the cell cycle should be analyzed.

We have performed western-blot and qPCR to check the *HPR1* expression and protein levels during G1, and have confirmed that both mRNA and protein are present throughout the cell cycle. We have included the new data in Figure S2d,e and in the text in page 7, line 241. Thanks

Minor comments

1. Beside Figure 1a, what are the experimental conditions used for auxin treatment (concentration, time of treatment)?

We have included this information in legend Fig.1 and methods (Page 15, lane 522; Page 21, line 812). Thanks

2. In Figure 1b, images are not representative as Rad52 foci seem more frequent in *sen1-aid* cells in contrast to the quantification. What does mean n? Number of experiments (and then how many cells per experiment) or number of cells?

We have included the requested details in legend Fig.1 in page 15, (lane 527). Thanks.

3. Figure 4 is far to be convincing. Adding more experiment would improve the statistical relevance of this experiment.

We have performed more experiments as suggested and included the data in the graph (Fig. 2b), so that the statistical relevance of the experiment is now clear. Thanks.

4. 25 out of 57 references (37%) correspond to self-citations. Although Aguilera's lab is undoubtedly a pioneer and leader in R-loop understanding, it is maybe exaggerated.

Sorry about that. It was not at all our intention to demerit the works of others. We have deleted references from our lab and added others from other labs to reduce the percentage mentioned to 25% without counting a couple of collaborations. Please, notice that some of the references of our lab, in any case, are reviews that consider the relevant work of other labs and those from our lab that are not cited here. In addition, many references refer to yeast work where the work of other authors is less abundant than from mammalian cells. We are happy to include additional references that referees consider necessary. Thanks

5. Methods section could be more detailed. For example, there is confusion between SC and SD, pronase treatment is not detailed...

We have extended the method details requested and included two tables with primers (Table S2) and plasmids (Table S3) used in the study. Changes are highlighted in blue on pages 20-22. Thanks

Reviewer #2 (Remarks to the Author):

The Aguilera has led the field in defining the effects of R-loops on genome stability using yeast as a model system. They previously defined factors that prevent (THO complex) and remove (Sen1 RNA-DNA helicase) R loops and showed that head on (HO) conflicts between transcription and replication (T-R) conflicts are more detrimental than are co-directional (CD) conflicts. The current study used an auxin-inducible degron (AID) system to examine the cell cycle dependent effects of Hpr1 (THO component) and Sen1 loss, with a focus on establishing whether R loops cause or are the result of T-R conflicts. Loss of either protein elevated R-loop accumulation (chromatin spreads and DRIP at GCN4 and PDR5), strongly stimulated recombination and caused delays in S-phase progression. A key experiment was the use of synchronized cells to demonstrate that loss of Hpr1 was associated with R-loop accumulation throughout the cell cycle (R loop formation does not require T-R conflicts), while loss of Sen1 was associated with R-loop accumulation in S phase. Metaplot analyses of genomic data (R loops by DRIPc-seq; H2AP and Rad52 by CHIP-seq) corroborated the distinct roles on Hpr1 and Sen1 on R-loop accumulation, with HO-CD difference in T-R conflicts being evident only in the *sen1*-AID strain. Altogether, the data convincingly demonstrate distinct and unexpected cell cycle-specific differences in R-loop accumulation and consequences upon loss of Hpr1 versus Sen1. The absence of THO results in co-transcriptional R-loops in G1 that cause subsequent replication fork blockage in S phase, while Sen1 loss only has effects in S phase and is associated with a HO bias for R loops as well as an asymmetry in damage. This is an important study that will fundamentally change how the field relates R loops to genetic instability.

Thanks very much for the positive reception of the manuscript and the constructive suggestions.

Major comments

1. The model needs a bit more explanation. What are the non-Sen1 helicases that remove co-transcriptional R-loops? How does this model explain the observed HO bias in yeast? Doesn't the model predict that the HO bias should only be evident in the absence of Sen1? Or, is there something that preferentially deals with CD conflicts that occur outside of S phase? Lines 351-352 imply that something is "more efficiently" removing R loops at CD collisions. Although not required, directly examining how the HO bias is affected by Hpr1 versus Sen1 loss would be a welcome experiment.

We have tried to clarify better the step of DNA-RNA removal by unwinding for the model (lines 485-487), but we prefer not to speculate at this point. Thus, there is several known co-transcriptional DNA-RNA helicases that could make the work, such as human UAP56/DDX39B and other DDX helicases reported, but their specific role in vivo has not been established, and certainly not for the yeast counterpart. We agree that in *sen1* cells, most R loops would form at HO collisions and this is, indeed, indicated in the explanation of the model in Discussion (lines 489-491). I am not sure about the question on how Sen1 or Hpr1 would affect HO bias, since our results indicate that the "prevention" function of THO, whether in G1 or S, and the S-specific "removal" function of Sen1, explain the HO bias. Certainly, the next step would be to whether specific factors deal with CD versus HO T-R conflicts; however, to do so we would need to define the factors involved in each case and the precise mechanism and DNA structures involved. These are very intriguing questions but belong to a long-term project beyond the scope of this manuscript. We thus would prefer not to get into a long and unprecise discussion. Thanks very much.

2. The English needs editing throughout.

We have tried to do so. We hope to have succeeded.

Minor comments

1. The authors need to define the THO complex components at its first mention. Defined as requested in page 3 (line 83). Thanks.

2. In Fig 1b the *sen1-AID/+aux/-RNH* panels for Rad52-YFP and DAPI are identical. They are different images. The first one only shows Rad52-YFP and the second shows the same +DAPI. The image for Rad51-YFP+DAPI could seem less stained for DAPI in this particular example. We have enhanced the brightness, in any case, to make it clearer.

3. Fig 1d – too many significant figures for -fold effects. We believe that fold change adds important information, as it is an intrinsically normalized value, and allows to realize the magnitude of the differences.

4. Fig S1c – why do the chromatin spreads (DAPI staining) look so different from WT? For the procedure of the chromosome spreading method, we break the cells and spread the DNA content on the slide. The nuclei of each sample could present different sizes and shapes due to the manual technique of this spreading step of the protocol.

5. Lines 152-156: there is no region 1 or 2 (should be 3' and 5'?) depicted in Fig 2b. Corrected as requested in page 6 (lines 212, & 216). Thanks

Also, the WT BrdU data should be shown for comparison in Fig 2b, especially since auxin has subtle effects on the cell cycle in Fig 2a.

We performed an initial kinetics with WT +/-aux and did not observed differences, therefore we continued using the -aux condition for each strain as the best control. We show this result here for the consultation of the referee, but we do not consider necessary to include it in the manuscript.

ChIP analysis of BrdU incorporation into the DNA at the genomic regions and amplicons depicted in the schemes (top) in the WT strain with and without auxin treatment.

6. Any idea why the BrdU signal increases in the proximal regions (relative to WT) in the *sen1-aid* strain? It is possible that this method extends the signal of the replication blocks, but it is more likely that the *Sen1* loss makes replication fork stalling more localized.

7. Line 165 – the “>16%” is misleading (implies only a 16% increase in the signal) and should be changed “from 4% to 16%” or “~ 4-fold.” Same comment for line 174.
Changed as suggested, in page 7 (lines 226 & 235).

8. Fig 4a – define again what TIR1 is.
Defined as requested in page 16 (line 561).

9. Line 197 – why “again” when hpr1-aid and sen1-aid were the same with respect to persistent Rad52 foci in G₁?
We did not analyze Rad52 foci in G₁, we normally count these foci in G₂ cells. We refer to the H2AP accumulation data in G₁.

10. Fig 5a – peak “enrichment” in text implies the amplification of pre-existing peaks. The data shown, however, do not convey an increase in peak signals but rather the appearance of new peaks.
We meant to say enrichment in peak number. We have modified it accordingly (page 8, line 285).
Thanks

11. Line 240 does not make sense.
We have rephrased it as suggested (page 9, line 303). Thanks

12. Why is there an enrichment of anti-sense as well as sense signals in Fig 5c?
This plot refers to the pool of genes as a population, and does not necessarily mean that the RNAs forming the hybrid are generated by the same gene that show sense and the anti-sense transcripts, or if so, whether they express both RNAs seq at the same time and in the same cell. This can be seen in the specific regions shown in Fig. 5a.

Reviewer #3 (Remarks to the Author):

Reviewer #3 (Remarks to the Author):

The paper by Martin-Alonso et al, aims at elucidating the mechanism(s) that prevent R-loop formation/accumulation at different cell cycle stages. They used hpr1 and sen1 conditional mutants, that are known to accumulate R-loops, and analysed R-loop accumulation at different cell cycle stages. They found that while hpr1 mutants accumulate R-loops in G₁ and S phase, Sen1 prevents R-loop accumulation, specifically in S phase. The most important message here is that R-loops can form co-transcriptionally, and independently of DNA replication.

Thanks very much for the positive reception of the manuscript and the constructive comments.

Specific Comments:

1. RnaseH overexpression in hpr1 and sen1 cells decreases recombination frequencies marginally but significantly (Fig1D). However, percentage of Nuclei with detectable hybrids decreases to wild type levels following RnaseH expression (Fig1E and also 1F). Question is, how much is recombination really depended on R-loops? Authors should discuss this point.

This is a very interesting question, indeed. Certainly, and as have been shown in other publications (Huertas and Aguilera, Mol Cell 2003; Mischo et al, Mol Cell 2011), not all recombination events are suppressed by RNase H overexpression. Since these mutations also affects either transcription elongation and termination part of the events may be associated with transcription stalling without the need of R loops. However, it is not possible to quantify this value, because the frequency of recombination refers to single events occurring in population of cells and i) percentage of cells with S9.6 signals refers to the whole genome and ii) not all hybrids cause recombination events, many would be resolved without resulting in recombination. We have tried to discuss and clarify this in the text (page 12).

2. Authors selected 2 regions which are believed to have head-on collisions between forks and

transcription and used BrdU-ChIP, to measured fork movement with respect to time. Due to differences in the replicates (3 qPCR triplicates), it is difficult to visualize the differences between +Aux and –Aux. Surely, when you take average you do see the differences, but is this significant?

We obtained the p value performing the Wilcoxon statistical analysis. We have included this information in the text (Page 15, lane 544)

3. (Brambati et al) showed that in sen1 mutants besides, ARS1211, another dormant origin is activated (ARS1211.5). This is ignored in the paper and should be commented. It is formally possible that higher signals seen in sen1 derive from the activation of ARS1211.5 (region 4).

This is an interesting point. We have included the reference as suggested, even though as explained (page 6) we see aberrant incorporation in region 3 but not in region 4 even at early time points, and a similar result is obtained in the other analyzed ARS, making it unlikely that it derives from ARS1211.5

4. The difference in H2A-P chip between –RNH and +RNH is not really significant at GCN4 but significant at PDC1. This needs a comment.

We have performed more experiments as suggested and included the data in the graph (Fig. 2b). As also indicated to referee #1, the significance is now clear. Thanks.

General comments:

The field of R-loop formation and accumulation is rather confusing. Particularly, it is generally believed that R-loops are always genotoxic and result from the uncoordinated clash between replication and transcription. This paper contributes to clarify an important point and convincingly demonstrate that R-loops can form independently of replication. Overall, the paper is solid and the results convincing. However, I strongly encourage the authors to clarify in the text that the R-loop concept is an interpretation as they monitor the presence of RNA-DNA hybrids, not R-loops. In fact, I would rather prefer that throughout the results section they refer to RNA-DNA hybrids and then, in the discussion, the clarify this issue. For instance, it is possible that in G1 the hybrids are in a R-loop conformation while in S phase they are not due to replication of the non transcribed strand.

We certainly agree with this point and are aware of the difference between R-loops and DNA-RNA hybrids, as indeed we have pointed out in our published reviews. We have made a clarification at the beginning of Results on why we use the term R-loop in most cases (lines 173-176) and we have discussed it further in Discussion (lines 424-428). We have also reduced the use of the term R-loop in Results.

With respect to the possibility that in S phase they are not R loops but rather just hybrids because of the replication of the non-transcribed strand, this is possible but unlikely. Transcription occurs on dsDNA, not on single-stranded template while the non-transcribed strand is being replicated. Therefore, it is likely that even in S phase, they are R loops. One way to have a DNA-RNA hybrid alone is if it is formed between the two sister chromatids or by uncoupling lagging-leading as previously proposed (Barroso et al EMBO Rep 2019; Svikovic et al, EMBO J 2019), but it is still speculative. In any case, we hope that the modifications added help clarify the differences. Thanks very much.

All Genomic data are in GEO. To review GEO accession GSE159870:

Go to <https://www.ncbi.nlm.nih.gov/geo/query/acc.cgi?acc=GSE159870>

Enter token ejijomwmtbwrjwz into the box.

REVIEWERS' COMMENTS

Reviewer #1 (Remarks to the Author):

In the revised version of their manuscript, Aguilera and colleagues answered very precisely and seriously to my comments and include many appropriate and convincing additional data. To my opinion, they perfectly addressed all concerns and this article now deserves publication in Nature Communications.

Reviewer #2 (Remarks to the Author):

The authors have adequately addressed issues raised in the initial review. This will be an important contribution to the literature.

Reviewer #3 (Remarks to the Author):

The authors have addressed the criticisms. The paper has been improved. Overall the results are relevant and convincing